# Lidocaine: A Local Anesthetic, Its Adverse Effects and Management

**DOI:** 10.3390/medicina57080782

**Published:** 2021-07-30

**Authors:** Entaz Bahar, Hyonok Yoon

**Affiliations:** Research Institute of Pharmaceutical Sciences, College of Pharmacy, Gyeongsang National University, Jinju 52828, Korea; entazbahar@gnu.ac.kr

**Keywords:** local anesthetics, lidocaine, adverse allergic responses

## Abstract

The most widely used medications in dentistry are local anesthetics (LA), especially lidocaine, and the number of recorded adverse allergic responses, particularly of hazardous responses, is quite low. However, allergic reactions can range from moderate to life-threatening, requiring rapid diagnosis and treatment. This article serves as a review to provide information on LA, their adverse reactions, causes, and management.

## 1. Introduction

A local anesthetic (LA) is a medicine that is used to numb a small part of the body temporarily before performing a minor surgery like skin biopsy. Before a dental operation, such as tooth extraction, LA may be given to the patient. LA do not cause humans to fall asleep, unlike general anesthesia. They are usually distinguished by their chemical structure, specifically the linkage between the compound’s common components, such as amide and esters [1]. The vast majority of the regularly used dental LA are amides, for example, lidocaine, mepivacaine, bupivacaine, etidocaine, prilocaine, and articaine [2]. The maximum recommended dose of LA may vary based on the nation, gender, age, weight, and medical condition of the patient. The gathering of information from several sources, Table 1 shows the acceptable maximum doses of LA with or without vasoconstrictor [3,4,5].

Lidocaine, also known as lignocaine, is a class Ib antidysrhythmic and local amino amide-based anesthetic that has been on the market since 1948 (Figure 1) [6,7]. Due to its superior safety profile as compared to other LA agents, it was quickly adopted. It can also be used to treat acute and chronic pain as an adjuvant analgesic [8,9,10]. It is widely used to relieve pain after a minor surgery or invasive procedures like biopsies, minor excisions, or dental surgery. However, as lidocaine can be used in different ways, i.e., by injection, inhalation, or as a topical agent to provide anesthesia to the same patients, it is essential to keep records of the total dose given to avoid its systemic toxicity. Lidocaine should not be used in patients with confirmed allergic hypersensitivity to amide-based LA. Due to the increase in the number of over-the-counter (OTC) drugs containing topical amide anesthetics such as lidocaine, the incidence of allergic contact dermatitis (ACD) to LA is rising. The ACD to LA is common, with an incidence of 2.4%, in which 32% of cases are linked to lidocaine [11]. The incidence of lidocaine allergy in 17 subjects out of 100 dentists (assumed patients) was detected, in which type I hypersensitivity was diagnosed in 13 cases, and four subjects had an IgE-independent allergy [12]. Sixteen cases of allergy contact dermatitis and delayed hypersensitivity to lidocaine were reported by Antoine et al. [13]. While local anesthetics have not been linked to serum enzyme elevation, they have been reported as potential causes of clinically evident liver injury when given as continuous infusions or repeated injections. Poisoning with the parenteral form of lidocaine is the most well-known, although poisoning with a topical spray formulation is also possible [14]. It has undesirable effects on the cardiovascular system (CVS) and the central nervous systems (CNS) once ingested in large amounts [15,16,17]. Toxicity to regional nerves and muscles is thought to be caused by long-term use of high drug concentrations, by the presence of preservatives in the amide-based LA solution such as lidocaine, or both [18,19]. During regional anesthesia, an inadvertent intravascular injection (primarily into the neck) of lidocaine causes severe cardiotoxicity such as hypotension, atrioventricular heart block, idioventricular rhythms, and life-threatening arrhythmias such as ventricular tachycardia and fibrillation, which are usually the first signs of LA toxicity [20]. A case of death of a 32-year-old male from a lethal dose of lidocaine was reported by Kalin et al. [21]. A case of death of a 76-year-old man with heart disease as a result of an excessive dose of lidocaine was reported [22]. A study regarding acute toxicity of lidocaine with a mortality rate of 10% was reported [7].

Antioxidants and preservatives in lidocaine, such as metabisulfite and parabens, may trigger allergic or adverse reactions in some people [23,24]. The most common allergic reaction is caused by the ester’s metabolic product, *para*-aminobenzoic acid, as cross-reactivity between esters is common [25]. By causing percutaneous and possibly ingestive sensitivity, parabens can cause allergic responses [26]. 

Parabens are a category of preservatives that is widely used in ointments, cosmetics, creams, lotions, dentifrices, toiletries, foods, and local anesthetics to inhibit the growth of microbes [27]. They are alkyl esters (methyl, ethyl, propyl, or butyl) of *p*-hydroxybenzoic acid, a chemical compound present in many fruits and plants that occurs naturally [28]. Its phenol-like activity, which probably works by protein denaturation and the antimetabolite properties of *p*-hydroxybenzoic acid, has shown bacteriostatic, fungistatic, and oxidant properties [29]. Chemical structures (Figure 2) and actions of the commonly used parabens are very similar, with the “R” group changing as shown in Table 2. The most widely used parabens are methylparaben, ethylparaben, propylparaben, and butylparaben, while many parabens (isopropyl-, isobutyl-, pentyl-, phenyl-, benzyl-) have been used, too.

The LA solutions typically contain 0.1% methylparaben, and the effective concentration is low (0.1–0.3%). Methylparaben is metabolized to *para*-aminobenzoic acid (PABA), which is a highly antigenic substance and is most likely a source of allergic reactions [30,31,32]. One of the most regularly used parabens, methylparaben, has been linked to T cell-mediated sensitivity, with urticarial maculopapular rash [33,34,35]. Microscopic examination of skin revealed mild-to-severe dermal inflammation and hyperkeratosis with acanthosis after 3 months of exposure to a product containing 0.2% methylparaben and 0.2% propylparaben in rabbits [26]. However, it remains poorly known whether parabens used in LA solutions are truly a source of allergic reactions.

Scientists have worked tirelessly to increase the efficacy and reduce the adverse reactions connected with lidocaine. Despite the fact that allergic reactions to lidocaine are quite rare, they can be true. Notably, patients who are allergic to lidocaine cause a challenge to the dentist in terms of delivering adequate treatment and managing postoperative pain [36]. Moreover, the acceptable limit for the incidence of true allergies to lidocaine is below 1%, so practitioners must be trained and educated properly in order to manage and diagnose a true LA allergic reaction [37]. Unfortunately, lack of awareness of adverse reactions to LA as well as the lack of allergy testing, diagnosis, and management has resulted in unavoidable dental consequences. Therefore, our review aims to provide informative descriptions regarding LA, their adverse reactions, causes, allergy testing, diagnosis, and management.

## 2. Identifying Allergic Reactions in Adverse Events

### 2.1. Fear or Anxiety-Related Adverse Reactions

Unintended intravenous administration of LA, poisonous overdose, sensitivity, and idiosyncrasy may all be misinterpreted as true allergic reactions [38]. Toxic side effects of local anesthetics are caused by either systemic exposure or a local pharmacologic effect [39]. Potential precipitating factors include irrational needle fears, chair posture, liver or kidney dysfunction, maximum prescribed doses, adequate safety precautions, and concurrent drug interactions. The easiest and most effective way for detecting risk factors that can lead to an adverse incident is to take a detailed medical history [40]. When adverse effects occur, the provider’s familiarity with the patient is critical as it allows for quick diagnosis and successful care [41].

### 2.2. Psychogenic Effects

In dentistry, anxiety plays a significant role. It has been reported that a substantial portion of the population in the United States is becoming more worried about dental care [42]. The most common adverse events seen in a dental office are psychogenic effects. These psychogenic responses are often misdiagnosed as allergic reactions due to their similarities.

### 2.3. Allergic Reactions

The common symptoms of allergic reactions include anaphylaxis, urticaria, edema, bronchospasm, unconsciousness, hyperventilation, nausea, vomiting, and changes in heart rate or blood pressure [43].

It is important to know the distinctions between allergic and psychogenic reactions so that patients get the treatment they need [44,45].

## 3. Response to Allergies

Originally, the immune response system of the body was thought to be solely protective; however, extreme allergic reactions’ dangerous potentials were gradually discovered. Hypersensitivities, also known as allergies, are incredibly active immune responses in which the immune system destroys tissue when fighting a possible risk, or an antigen, which would otherwise be safe to the individual. A case of sudden death after a gingival injection of lidocaine was reported, with suspected overdosing or anaphylactic shock [46]. These reactions can range in severity from mild to life-threatening, and the clinical manifestations of an antigen reaction can range from mild (with minor skin manifestations over time) to those requiring immediate diagnosis and aggressive treatment to avoid respiratory and cardiovascular collapse which can lead to death. An allergic reaction occurred 30 min after a local infusion of lidocaine for the retraction of retained teeth in an 86-year-old woman [47]. There are several forms of hypersensitivity reactions, which are better categorized based on the disease’s immunologic mechanism (Table 3).

## 4. Allergy Testing Procedures

A detailed account of the incidents as well as a thorough review of the history of a recorded allergic reaction is needed. The drugs used, the onset of the reaction, signs and symptoms, and the duration of the outbreak are all essential factors to consider when diagnosing a true allergic reaction. The majority of the reported adverse reactions are psychological, with only a small percentage caused by an avoidable intravascular injection. It is important not to label a patient as allergic too quickly; instead, the true nature of the problem should be investigated. If the reaction is serious and clearly indicates an allergic reaction, a referral to an allergist is considered standard of care [50]. To help in the selection of a safe local anesthetic for a particular patient, allergists use skin prick tests (SPTs), intradermal or subcutaneous positioning tests, and/or drug provocative challenge testing (DPT) procedures. 

Typically, an SPT is conducted, which involves softly pricking the skin with a plastic applicator to inject a small quantity of an LA solution. The arm is used for this test, and a red raised itchy hive emerges on the skin within 15–20 min due to LA sensitivity [29]. If an allergic reaction takes place, the required allergic reaction treatment protocol must be implemented. If the test is carried out with a highly diluted agent and the results are negative, a more concentrated agent could be used [51]. If the SPT is negative, an intracutaneous or intradermal test, in which a small amount of the test solution is injected into the epidermis of the forearm and the site is examined for 20 min for wheal or flare reactions, is sufficient. Subcutaneous provocation testing begins with 0.1 mL of the undiluted local anesthetic solution followed by 0.2, 0.5, 1.0, and 2.0 mL into the extensor side of the patient’s upper arm at 30-min intervals if the prick and intradermal tests are negative [52]. Only if the case history, skin examination, and the laboratory test yield ambiguous results, DPT with the substance in question is performed [53]. Many allergists regard DPT as the gold standard in the diagnosis of drug allergies; nevertheless, there is concern about the test’s potential side effects [54]. Before beginning any DPT, an individual risk-benefit analysis should be completed, and strict surveillance with emergency protocols should be implemented. In general, the clinician should start with a low dose and gradually increase it, discontinuing administration as soon as any signs or symptoms arise [54]. The effectiveness of this procedure is dependent on the extremely rare occurrence of a true allergic reaction to amide-based local anesthetics; however, the testing relieves stress for both the patient and the doctor, and it may enable diagnosis of the extremely rare amide allergy [54].

Unfortunately, there is no reliable in vitro allergy screening procedure that can be used on a regular basis. Gall et al. used a self-made radioallergosorbent test with polystyrene discs and a local anesthetic, but all of the patients were negative [55].

## 5. Management of Adverse Reactions

### 5.1. Immediate Management

#### 5.1.1. General Considerations

Local anesthetic systemic toxicity (LAST) is an essential side effect to be aware of, ranging from minor symptoms to serious cardiac or central nervous system (CNS) problems. The following guidelines should be followed in general for the management of toxic reaction of LA [3].

(a)Treatment should be customized for the affected person.(b)The patient should be lie in the supine position, with the face and torso facing up, with extended legs, and injuries should be avoided.(c)Basic life support, ABCs (airway, breathing, and circulation/compression) should be supplied as required.(d)If a seizure lasts more than a few minutes, oxygen should be given.(e)In case of persistent seizures, an effective anticonvulsant should be explored, for example, a benzodiazepine, diazepam, thiopental, etc.(f)Adequate observation followed by management of the signs and symptoms as required (such as hypotension, apnea, and cardiovascular collapse).

The first line of defense against LAST should be airway management, circulatory support, and avoiding systemic side events. Quick breathing and oxygenation can help with resuscitation and reduce the danger of seizures and cardiovascular collapse. LAST is addressed symptomatically with pharmacologic therapies such as benzodiazepines, barbiturates, or propofol, which raise the seizure threshold. Hyperventilation (high-dose oxygen) reduces the cerebral blood flow and has been used to improve the seizure threshold [56,57]. 

The introduction of lipid emulsions reduces the plasma concentration of free accessible LA, which is the other premise of treatment. The infusion of lipid emulsions binds free circulating local anesthetics and lowers plasma levels due to their high lipid solubility. The American Society of Regional Anesthesia and Pain Medicine (ASRA) publish its management recommendations concerning LAST on a regular basis to reflect new information, user input, and simulation [58,59,60,61,62]. Figure 3 reflects a part of a clinical system for managing LAST suggested by the American Society of Regional Anesthesia and Pain Medicine (ASRAPM).

Successful outcomes have been reported even after prolonged resuscitation, which may be explained in part by suggestions in animal models that bupivacaine, when added to a cardioplegia solution, actually improves the function and reduces cellular damage of isolated rat hearts after prolonged cold storage.

#### 5.1.2. Pharmacotherapy Management of Anaphylaxis/Anaphylactoid Reactions

Epinephrine is the primary and first medicine of choice in the treatment of anaphylaxis because it has the ability to sustain blood pressure while also relaxing bronchial smooth muscles. Furthermore, epinephrine efficiently counteracts the negative effects of circulating mediators [63]. In anaphylaxis, there is no known dosage or regimen for intravenous (IV) epinephrine. Due to the risk of potentially fatal arrhythmias, epinephrine should only be given IV during cardiac arrest or to profoundly hypotensive patients who have failed to respond to IV volume replacement and many intramuscular (IM) epinephrine injections (Table 4) [48].

### 5.2. Preparations without Preservatives

Patients that are resistant to ester-based local anesthetics should be treated with a preservative-free amide-based local anesthetic, whether based on medical history or intradermal skin testing. To prevent an allergic reaction to the PABA metabolite of methylparaben, a preservative agent, an amide-based local anesthetic without preservatives should be selected.

### 5.3. Antihistamines

Antihistamines have a chemical relationship with caine-type local anesthetics, which may clarify how they function as local anesthetics. Rosenthal and Minard discovered in 1939 that diphenhydramine induced local anesthesia that was equivalent to that produced by 1% procaine [67,68]. Despite the manufacturer’s warning against using diphenhydramine as a local anesthetic, multiple reports of its usage in dental and minor surgical procedures have appeared in the literature since then [67,68,69,70]. Diphenhydramine has a longer onset and shorter time of action than lidocaine. With diphenhydramine, fewer patients report achieving complete anesthesia [67,68,69,70]. 

### 5.4. Epinephrine

Epinephrine is an alpha/beta agonist that is used in LA cartridges as an adjuvant. Epinephrine is also used as a first-aid medication for anaphylaxis and as a vasoconstrictor to reduce systemic absorption of LA and prolong the duration of anesthetic activity. Table 5 presents the formulations of LA containing epinephrine available in cartridges.

### 5.5. General Analgesia and Hypnosis

In patients who have hypersensitivity reactions to local anesthetics, general analgesia, such as inhaled nitrous oxide (N20), is an option. For certain patients, intravenous opioids may provide adequate analgesia during labor. In case of potential hypersensitivity to LA and in patients who have autonomic responses to local anesthetic administration, hypnosis is particularly useful [72].

## 6. Conclusions

True allergic reactions to LA are rare adverse events with unexpected outcomes, but effective therapy can save a patient’s life. If a probable allergic reaction occurs, the dentist must assess the events that have led up to the reaction and make a treatment plan. For proper diagnosis, the dentist must follow scientific guidelines for the management of allergic reactions discussed in this minireview. 

## Figures and Tables

**Figure 1 medicina-57-00782-f001:**
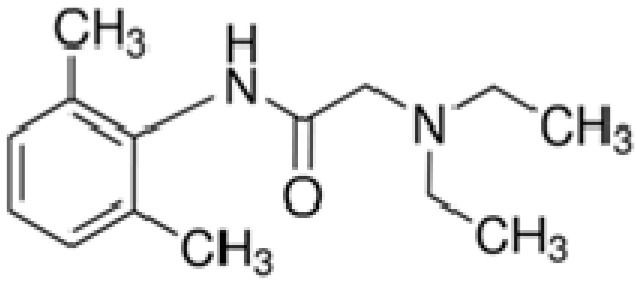
Chemical structure of lidocaine.

**Figure 2 medicina-57-00782-f002:**
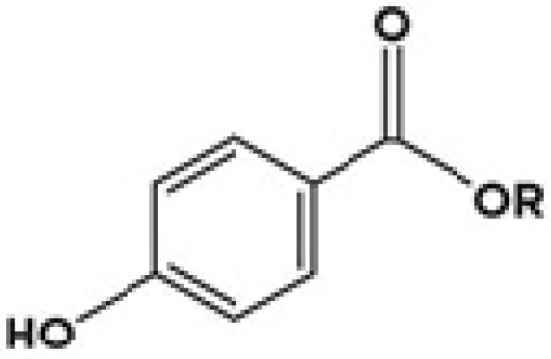
Common chemical structure of parabens.

**Figure 3 medicina-57-00782-f003:**
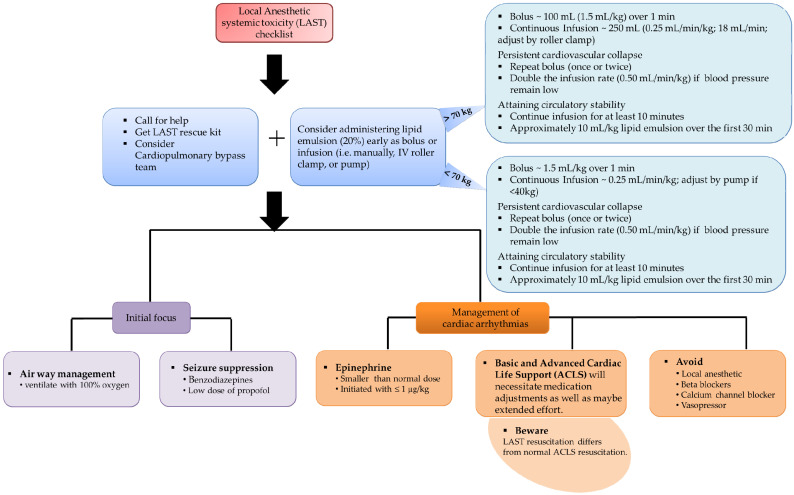
The American Society of Regional Anesthesia and Pain Medicine (ASRAPM) guidelines for the management of local anesthetic systemic toxicity (LAST) [59,60].

**Table 1 medicina-57-00782-t001:** Acceptable maximum dosages of commonly used local anesthetics [2,3,4,5].

Local Anesthetics (LA)	Concentration Available (%)	Maximum Recommended Dose (mg/kg)				* Maximum Recommended Total Dosage (mg)	
		Without Vasoconstrictor		With Vasoconstrictor		Adult	Children
		Adult	Children	Adult	Children		
Lidocaine	2.0	N/A	4.4	7.0	4.4	500	300
Mepivacaine	2.0	N/A	4.4	6.6	4.4	400	300
	3.0	6.6	4.4	N/A	4.4		
Bupivacaine	0.5	N/A	N/A	2.0	1.3	175	90
Etidocaine	1.5	4.5	N/A	6.5	N/A	400	300
Prilocaine	4.0	8.8	6.0	8.8	6.0	600	400
Articaine	4.0	N/A	N/A	7.0	7.0	500	500

*: maximum total dosage may need adjustment based on weight, age and medically compromised patients; %: percentage; mg: milligram; kg: kilogram; N/A: not applicable.

**Table 2 medicina-57-00782-t002:** Chemical structure of different parabens.

Substance	R
*Para*-hydroxybenzoic acid	-H
Methylparaben	-CH_3_
Ethylparaben	-CH_2_CH_3_
Propylparaben	-CH_2_CH_2_CH_3_
Butylparaben	-CH_2_CH_2_ CH_2_CH_3_
Benzylparaben	-CH_2_ C_6_H_5_

**Table 3 medicina-57-00782-t003:** Several forms of hypersensitivity reactions accelerated by local anesthetics and their management [48,49].

Hypersensitivity Reaction	Mechanism	Associated Disorder	Signs and Symptoms	Management
Mild allergy	Bodily histamine release response	Skin rash not associated with respiratory or cardiovascular issues	Itching, hives, and/or rash	Administration of a histamine blocker such as diphenhydramine by the intramuscular (IM), intravenous (IV), or oral route
Anaphylactic	Increased vascular permeability, edema, and smooth muscle hyperreactivity are all caused by IgE-sensitized mast cell mediators	Anaphylaxis, bronchial asthma, urticaria	A mild-to-moderate rash, erythema, or urticaria on the skin, swelling of the airways, erythema, pruritus, and edema, with or without angioedema, hypotension, tachycardia, dyspnea, gastrointestinal disturbances, severe bronchospasm, cardiac dysrhythmias, and cardiovascular collapse	Early administration of epinephrine (IM), maintenance of the airways and ventilation with 100% oxygen, positive pressure ventilation via a bag-valve-mask device, advanced airway adjuncts (e.g., supraglottic airways, endotracheal)
Anaphylactoid	Triggers the release of a combination of biochemical mediators, such as histamine, neutral proteases, prostaglandins, leukotrienes, and other chemokines and cytokines			

**Table 4 medicina-57-00782-t004:** Pharmacotherapy management of anaphylaxis/anaphylactoid reactions [48,64,65,66].

Treatment	Medications		Dose (mg/kg)	Route of Administration	Site of Administration
Primary treatment	Epinephrine		0.3	IM	Deltoid or vastus lateralis
			0.05–0.2	IV	Blood
Secondary treatment	Bronchodilator (β2-agonist)	Albuterol	0.09	Inhalation	Nasal
	H1-blocker (antihistamine)	Diphenhydramine	0.5	IV	Blood
	Optional				
	H2-blocker	Famotidine (Pepcid)	20	IV	Blood
	Steroids	Hydrocortisone	1–2.5	IV	Blood
		Methylprednisolone	1	IV	Blood

H: histamine; IM: intramuscular; IV: intravenous.

**Table 5 medicina-57-00782-t005:** Ratio of local anesthetics (LA) and epinephrine available in cartridges [2,4,5,71].

Local Anesthetics (LA)	Formulation (LA: Epinephrine)		Maximum Recommended Dose (mg/kg)		* Maximum Recommended Total Dosage (mg)	
	Adult	Children	Adult	Children	Adult	Children
Lidocaine, 2%	1:50,000, 1:100,000	1:50,000, 1:100,000	7	4.4	500	300
Bupivacaine, 0.5%	1:200,000	1:200,000	2	1.3	175	90
Prilocaine, 4%	1:200,000	1:200,000	8.8	6	600	400
Articaine, 4%	1:100,000, 1:200,000	1:100,000	7	7	500	500

*: maximum total dosage may need adjustment based on weight, age and medically compromised patients; LA: local anesthetic; %: percentage; mg: milligram; kg: kilogram.

## Data Availability

The data presented in this study are available in this article in Medicina.

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
