# Peer review of "Lidocaine: A Local Anesthetic, Its Adverse Effects and Management"

_medicina, 2021, doi:10.3390/medicina57080782_

Round 1
Reviewer 1 Report
This article is a brief review on the subject of local anaesthetics. Overall, it is well organised, informative, and easy to consult for clinical practice. The bibliography is sufficiently comprehensive and up-to-date. However, the purpose of the article is never stated. Apart from the abstract, it is never stated that this is a review and, above all, what the specific purpose of this review is, what makes it unique. The authors should explode this aspects in the 'Introduction' paragraph.
Author Response
This article is a brief review on the subject of local anaesthetics. Overall, it is well organised, informative, and easy to consult for clinical practice. The bibliography is sufficiently comprehensive and up-to-date. However, the purpose of the article is never stated. Apart from the abstract, it is never stated that this is a review and, above all, what the specific purpose of this review is, what makes it unique. The authors should explode this aspects in the 'Introduction' paragraph.
Reply: Thank you very much for your valuable comments and suggestions. We included a paragraph in the introduction section to detonate the purpose of the present review. Please see the red tract change text in the introduction section. Line 91-109
Reviewer 2 Report
Thank you for your work!
Dear collegues there are some issues that need corrections.
The article as a review needs some more focus on lidocaine as adverse effects, central effects (i.e. after an inadvertent intravascular injection) and management.
In my opinion you gave too much space for description of allergic tests.
Please describe the managment and resuscitation guidelines as guidelines of an international assotiation (ASA, ERC, ASCIA or similar), maybe with a diagram.
Epinephrine is not an antidote as stated in Section 5.4, because antidotes have precise charateristics.
Please include the incidence of allergic reactions.
Author Response
The article as a review needs some more focus on lidocaine as adverse effects, central effects (i.e. after an inadvertent intravascular injection) and management.
Reply: Thank you very much for your valuable comments and suggestions. We included more details about lidocaine systemic toxicity and management as per the reviewer suggestions. Please see the red tract change text in sections 1 (Introduction, line 29-40; 46-52) and 5 (Management of adverse reaction, line 205-237).
In my opinion you gave too much space for description of allergic tests.
Reply: We have modified the description of allergic tests to make it more concise. Please see section 4. Line 152-200
Please describe the managment and resuscitation guidelines as guidelines of an international assotiation (ASA, ERC, ASCIA or similar), maybe with a diagram.
Reply: Thank you for your suggestions. We included the guidelines for the management of Local anesthetic systemic toxicity suggested by The American Society of Regional Anesthesia and Pain Medicine (ASRAPM). Please see the red tract change text in sections 5.1.1 and Figure 3. Line 218-238
Epinephrine is not an antidote as stated in Section 5.4, because antidotes have precise charateristics.
Reply: We have revised the sentence as “Epinephrine is also used as a first-aid medication of anaphylaxis”….
Please include the incidence of allergic reactions.
Reply: We have included some incidences of lidocaine allergic reactions in the introduction section of the Lidocaine paragraph. Please see the red tract change text in the introduction part. Line 29-56
Round 2
Reviewer 2 Report
Dear collegues,
I have no more suggestions.
Thank You!